# The aroma of TEMED as an activation and stabilizing signal for the antibacterial enzyme HEWL

Zahra Seraj[1,2], Shahin Ahmadian[1], Matthew R. Groves[2]*, Arefeh Seyedarabi [1]*

1 Department of Biochemistry, Institute of Biochemistry and Biophysics, University of Tehran, Tehran, Iran,
2 Department of Drug Design, University of Groningen, Groningen, The Netherlands

* a.seyedarabi@ut.ac.ir (AS); m.r.groves@rug.nl (MRG)

**Data Availability Statement:** All relevant data are within the paper and its Supporting Information files.

**Funding:** We acknowledge financial support of the Presidential Deputy for Science and Technology,

## Abstract

The unpleasant smell released from dead bodies, may serve as an alarm for avoiding certain behaviour or as feeding or oviposition attractants for animals. However, little is known about their effect on the structure and function of proteins. Previously, we reported that using the aroma form of TEMED (a diamine), representative of the "smell of death", could completely inhibit the fibril formation of HEWL, as an antibacterial enzyme, and a model protein for fibrillation studies. To take this further, in this study we investigated the kinetics of TEMED using a number of techniques and in particular X-ray crystallography to identify the binding site(s) of TEMED and search for hotspot(s) necessary to inhibit fibril formation of HEWL. Structural data, coupled with other experimental data reported in this study, revealed that TEMED completely inhibited fibril formation and stabilized the structure of HEWL through enhancement of the CH-Π interaction and binding to an inhibitor hotspot comprised of residues Lys33, Phe34, Glu35 and Asn37 of HEWL. Additionally, results from this study showed that the binding of TEMED increased the activity and thermal stability of HEWL, helping to improve the function of this antibacterial enzyme. In conclusion, the role of the "smell of death", as an important signal molecule affecting the activity and stability of HEWL was greatly highlighted, suggesting that aroma producing small molecules can be signals for structural and functional changes in proteins.

## Introduction

Sense of smell in animals and human beings help them to find food, detect dangerous situations and provide cognitive influences, especially in human beings. These smells or aromas, arise from the properties of some compounds, which are found in their gaseous phase due to their small size and low vapour pressure.

Researchers have looked at aroma producing compounds from different points of views. Some researchers have investigated the effectiveness of the compounds on activation of some signaling pathways in different parts of the body, which may or may not even involve the olfactory system (such as the influence of the smell of sandal wood on hair growth [1]). On the

the Institute for Research in Fundamental Sciences and the Iranian Light Source Facility for the opportunity to be users at the ALBA synchrotron program 2017 (proposal number ID2017062245). We acknowledge financial support of the OPEN SESAME for the two months student training fellowship for Miss Zahra Seraj (2018) and the Ministry of Science and Technology of Iran for six months research placement of Miss Zahra Seraj at University of Groningen, Netherlands.

**Competing interests:** The authors have declared that no competing interests exist.

**Abbreviations: ACT**, Acetate; **DSF**, Differential scanning fluorimetry; **EDO**, Ethylene glycol; **Not-heated**, Not-heated HEWL, at room temperature; **Not-treated 5h**, HEWL incubated for 5 hours at 54 ℃ but Not-treated; **Not-treated 24h**, HEWL incubated for 24 hours at 54 ℃ but Not-treated; **PGO**, 1,2 Propanediol; **TEMED**, N,N,N,N'-Tetramethylethylenediamine; **TEMED5h**, HEWL treated with TEMED for 5 hours at 54 ℃; **TEMED24h**, HEWL treated with TEMED for 24 hours at 54 ℃.

other hand, some have studied the psychological or physiological effects of aroma producing compounds on human beings or other animals [2]. For instance, Hussain *et al.*, in 2016, demonstrated that flies appear to use their sense of smell as a signal to find polyamine-rich foods and that polyamine-rich diets significantly increased the number of progenies of flies [3]. Polyamines are a class of aroma producing small molecules that can be generated via a number of ways including endogenous biosynthesis; microbes in the gut [4]; and the action of polyamine-synthesis enzymes (i.e. ornithine decarboxylase aids decarboxylation of amino acids forming polyamines such as putrescence and cadaverine [3]), which have bacterial origin and make the "smell of death" from the bodies of dead animals. The smell of polyamines can be detected by animals and at higher concentrations are unpleasant to human beings. It is interesting that both the deficiency and excess of polyamines can be deleterious to health and reproduction [5].

In our previous study [6], we investigated the effect of three different aroma producing small molecules including two polyphenols with pleasant smells (Cinnamaldehyde and Phenyl ethyl alcohol) and a polyamine with an unpleasant smell, (N,N,N,N'-Tetramethylethylenediamine/ TEMED), on the fibrillation process of Hen egg white lysozyme (HEWL), as a model protein in fibrillation studies. Based on the results achieved in that study, it was clear that TEMED (a representative of polyamines), completely stopped the process of fibril formation in HEWL (an antibacterial enzyme that lyses the cell wall of bacteria) and improved its enzymatic activity. To take that further, it was interesting for us to investigate whether compounds in their aroma form can impact the thermal or structural stability of proteins and affect their function. Therefore, in this study, we investigated the fibrillation kinetics of HEWL in the presence of TEMED, using a number of techniques including ThT and intrinsic fluorescence spectroscopy, Circular Dichroism spectroscopy, Atomic Force microscopy, Dynamic Light Scattering, SDS-PAGE, and in particular X-ray crystallography, to reveal the binding mode and mechanism for the prevention of fibril formation in HEWL. Differential scanning fluorimetry was also used to investigate the thermal stability of HEWL in the presence of TEMED in the aroma form and at different concentrations in solution. Our results showed that TEMED behaves as a specific ligand for HEWL, binds to the active site of HEWL, inhibits fibrillation and increases the activity and thermal stability of the enzyme.

## Materials and methods

### Materials

Hen egg-white lysozyme or HEWL (catalogue number L6876), Thioflavin T or ThT, Nile red, glycine (CAS No. 56-40-6), Sodium dodecyl sulphate (SDS) (Catalog no. 85,192–2), N,N,N,N'-Tetramethylethylenediamine or TEMED (CAS No. 110–189), and sodium acetate (CH3COONa) (lot 110H-072015) were all purchased from Sigma-Aldrich. Protein gel marker (PM1500) was purchased from SMOBiO. Mica for atomic force microscopy (ca. 92680) was purchased from PELCO. Acrylamide (UN-NO 2074), Amicon Pro-Affinity Concentration Kit Protein G (ACK5010PG) and N,N'methylendiacrylamid (EC.NO. 203-750-9) were purchased from Merck. NaCl (Lot 24091) was purchased from SERVA. PEG400, ethylene glycol and 1,2 Propanediol were from Molecular dimensions CryoProtX MD$_1$-61. Hampton Research Crystal Screen II for crystal growth (CAT NO. HR2-112), 24 well crystallization VDX Plates with or without sealant (CAT NO. HR3-172 and HR3-142, respectively), OptiClear Plastic Cover Slides (CAT NO. HR8-074) and Siliconized Glass Cover Slides (CAT NO. HR3-239) were all purchased from the Hampton Research company. High vacuum grease was purchased from Girovac. For DSF experiments, Hen egg-white lysozyme or HEWL (Lot number P02C037) and Sypro Orange (catalogue number S6650) were purchased from Thermo fisher. Glycine

(Lot number AO359169) was purchased from ACROS. N,N,N,N'-Tetramethylethylenedia-mine or TEMED (CAS number 110-18-9) and PCR plates (catalogue number 82006–636) were purchased from VWR.

## HEWL sample solution preparation and incubation studies

2 mg/ml (140 μM) HEWL solution was prepared in 50 mM glycine buffer pH 2.2. Sample TEMED5h and that of TEMED24h and Not-treated24h were incubated at 54 ˚C and 150 rpm for 5h and 24h, respectively [6], in either the presence or absence of TEMED, in the aroma form.

## Thioflavin T (ThT) fluorescence assay

HEWL samples incubated with or without aroma for 5h and 24h were diluted 50-fold with ThT solution (at 25 μM) and fluorescence intensities were recorded at 484 nm after excitation at 440 nm. Excitation and emission slit widths were both set at 5 nm. A sample of Not-heated HEWL was also used and ThT recorded as control. The results were repeated and standard deviation bar calculated for the graph using multiple data.

## Circular dichroism spectroscopy

Circular dichroism (CD) spectra of HEWL samples were recorded from 250 to 195 nm using an AVIV 215 spectrophotometer (Aviv Associates, Lakewood, NJ, USA). The sample prepara-tions were the same as described before in our previous study [6]. Three scans of each sample were measured and averaged. The control buffer scans were run in duplicate, averaged and then subtracted from the sample spectra. The results were plotted as ellipticity (deg. cm$^2$ dmol$^{-1}$) versus wavelength (nm).

## Protein gel electrophoresis

Tris-glycine SDS polyacrylamide gel electrophoresis (SDS-PAGE) was used under reducing conditions to analyze the HEWL samples in this study. A pre-stained protein marker was used.

## Atomic Force microscopy (AFM)

AFM scans were performed using a Veeco AFM instrument (Sharif University, Tehran, Iran). The sample preparation method used was the same as described before [6].

## Dynamic light scattering (DLS)

Dynamic light scattering measurements were performed using the Malvern Zeta Sizer Nano ZS. The apparatus and parameters used were the same as we described in our previous study [6].

## Intrinsic fluorescence intensity assay

Not-heated HEWL, treated- and Not-treated HEWL samples were diluted 25 times with 50 mM glycine pH 2.2. The excitation wavelength was 280 nm and emission spectra were recorded between 300 and 400 nm. Excitation and emission slit widths were both set at 10 nm.

## Differential scanning fluorimetry (DSF)

Stock solution (5000x) of the fluorescent dye Sypro Orange (Molecular Probes) was diluted with ultrapure water to a final concentration of 25x. The initial concentration of HEWL was 140 μM. Reaction mixtures were prepared by diluting HEWL in the presence or absence of desired concentrations of small molecules and 5 μL of 25x Sypro Orange, in order to yield a final reaction volume of 50 μl.

The thermal stability was examined for all samples and 15 μM concentrations of HEWL were used. DSF experiments were performed in a BioRad CFX96 RT-PCR machine programmed to the temperature range 20–90˚C and heating rate 1˚C / 0.5 min. All obtained curves were inspected manually to check their quality, and the $T_m$ values were determined using the first derivative curve. All experiments were performed in triplicate [7].

## TEMED-HEWL sample preparation and incubation studies using DSF

2 mg/ml (140 μM) HEWL solution was prepared in 50 mM glycine buffer pH 2.2. Sample TEMED5h and that of TEMED24h and Not-treated24h were prepared by incubation of HEWL at 54 ˚C and 150 rpm for 5h and 24h, respectively [6], in either the presence or absence of TEMED as indicated. In addition, a second series of samples was prepared in which HEWL was incubated at room temperature for 24h in the presence or absence of TEMED in solution.

To determine the TEMED concentration range to be used in solution, the experimental set-up initially used for detecting the effect of TEMED in the aroma form on HEWL fibrillation was considered. Therefore, if the vial containing aroma producing TEMED and the bottle containing HEWL were considered as two different systems, similar to the vapour diffusion method in crystallisation, there would be an equilibrium between the water (as a basic solvent of buffer) in the HEWL solution and TEMED in the aroma form circulating out from the vial. Therefore, since the initial volume of TEMED in solution was 50 μl, calculation was done to find the final concentration based on the reduced volume at the end of the incubation (S1 Table).

## Crystallisation

Crystals of Not-heated (or native) HEWL was obtained and used as the control in two different solutions consisting 50 mM glycine pH 2.2 and 50 mM glycine pH 8.6. In the case of treated HEWL samples (at 2 mg/ml incubated with aroma form of TEMED), samples were prepared as mentioned in our previous study [6]. For the purpose of crystallising the treated HEWL samples (TEMED5h and TEMED24h), the initial concentration at 2 mg/ml was increased to between 6 and 8 mg/ml, respectively, using a concentrator with 10,000 Da MWCO. For preparation of co-crystallised sample, HEWL was mixed with TEMED and incubated for 45 min at 4 ˚C (the molar ratio of TEMED was approximately 100 times more than HEWL), and then centrifuged for 5 min at 14000 rpm and used in crystallisation. The final concentration of co-crystallised HEWL was 15 mg/ml. Conditions 1 (2 M NaCl and 10% PEG 6000) and 9 (2 M NaCl and 0.1 M sodium acetate pH 4.6) from Hampton Research Crystal Screen II were used for crystallisation of Not-heated, treated and co-crystallised HEWL samples. The final cryo-protectant solutions were generally composed of the crystallisation conditions in which the crystals were grown in (with about 20% increase in precipitant concentration) and the addition of 25% (v/v) ethylene glycol (EDO) or 20% (v/v) 1,2 Propanediol (PGO). All crystals were obtained using the hanging drop method.

**Table 1. Data collection and refinement statistics.**

| Sample Name | pH = 8.6 | TEMED 5h | TEMED 24h | TEMED-co |
|---|---|---|---|---|
| PDB ID | 6ABN | 6AEA | 6AD5 | 6ADF |
| Space group | P43212 | P43212 | P43212 | P43212 |
| a, b, c (Å) | a = 78.17 | a = 79.46 | a = 78.94 | a = 78.47 |
| | b = 78.17 | b = 79.46 | b = 78.94 | b = 78.47 |
| | c = 36.97 | c = 36.95 | c = 36.97 | c = 37.02 |
| α, β, γ (°) | α = β = γ = 90 | α = β = γ = 90 | α = β = γ = 90 | α = β = γ = 90 |
| Resolution (Å) | 55.27–1.17 (1.23–1.17)[a] | 56.19–1.4 (1.48–1.4)[a] | 55.82–1.75 (1.84–1.75)[a] | 39.24–1.08 (1.13–1.08)[a] |
| Total number of observations | 423585 (57240)[a] | 205422 (28451)[a] | 104597 (14869)[a] | 568308 (76287)[a] |
| Total number unique | 39339 (5648)[a] | 23907 (3437)[a] | 12277 (1724)[a] | 50672 (7269)[a] |
| Multiplicity | 10.8 (10.1)[a] | 8.6 (8.3)[a] | 8.5 (8.6)[a] | 11.2 (10.5)[a] |
| Completeness (%) | 100 (100)[a] | 99.9 (99.5)[a] | 99.8 (99.5)[a] | 100 (100)[a] |
| Rsym [b] (Rmerge) | 0.209 (0.33)[a] | 0.14 (0.42)[a] | 0.111 (0.412)[a] | 0.067 (0.224)[a] |
| Mean I/Sigma (I) | 13.1 (5.8)[a] | 8.6 (3.8)[a] | 15.6 (8.6)[a] | 19.5 (8.4)[a] |
| Rpim [c] | 0.067 (0.109)[a] | 0.052 (0.150)[a] | 0.039 (0.144)[a] | 0.021 (0.072)[a] |
| Rmeas [d] | 0.22 (0.348)[a] | 0.151 (0.447)[a] | 0.118 (0.437)[a] | 0.7 (0.235)[a] |
| Resolution (Å) | 1.17 | 1.4 | 1.75 | 1.08 |
| Mosaicity | 0.28 | 0.41 | 0.54 | 0.39 |
| rmsd bond length (Å)/angle (°) | 0.031/2.461 | 0.028/2.543 | 0.023/2.182 | 0.032/2.675 |
| Mean B value (Å²) | 9.562 | 17.659 | 16.299 | 12.572 |
| R-factor/R-free (%)[e] | 16.48/19.56 | 16.19/19.79 | 13.79/18.04 | 15.21/17.55 |

[a] The parameter values for higher resolution are given in parentheses

[b] Rsym = $\Sigma_{hkl} \Sigma_I |I_i - |/\Sigma_{hkl} \Sigma I_i$, $I^i$ is the intensity of the i[th] observation, $<I>$ is the mean intensity of the reflection, and the summations extend over all unique reflections (hkl) and all equivalents (i), respectively.

[c] Rpim is a measure of the quality of the data after averaging the multiple measurements.

[d] Rmeas (also known as Rrim) is an improved version of the traditional Rmerge (Rsym) and measures how well the different observations agree.

[e] *R-factor* = $\Sigma_{hkl} |Fo\_Fc|/\Sigma_{hkl} Fo$, where Fo and Fc represent the observed and calculated structure factors, respectively. The R-factor is calculated using 95% of the data included in refinement and R-free the 5% excluded. The values presented in this Table come from SCALA [14] and REFMAC [20] from the CCP4 suite [16].

### Data collection and structure determination

Diffraction data were collected at the ALBA synchrotron source, the XALOC Beamline, at 100 K and at a wavelength of 0.9792 Å. The highest-resolution crystals diffracted to 1.08 Å. iMOSFLM [8] was used for data reduction, and Scala [9] was used for scaling and merging intensities. The structures were determined by molecular replacement using Phaser [10] with 1DPX as the search model. Refinement of the structures was performed using REFMAC version 5.8.0135 [20]. Once the structures were solved (Table 1), Ligplot+ [11], QtMG from CCP4 software [20], and Chimera [12] were used for a detailed structural analysis of the ligand binding site(s).

## Results and discussion

### Fibrillation kinetics

The main goal of this study was to investigate the kinetics of TEMED (as a representative of smell of death) in inhibiting fibril formation of HEWL, as an antibacterial enzyme. Previously we reported that incubating HEWL at 54 ˚C in 50 mM glycine pH 2.2 for 24h in the presence of TEMED in its aroma form prevented fibril formation, while in the absence of TEMED, fibril

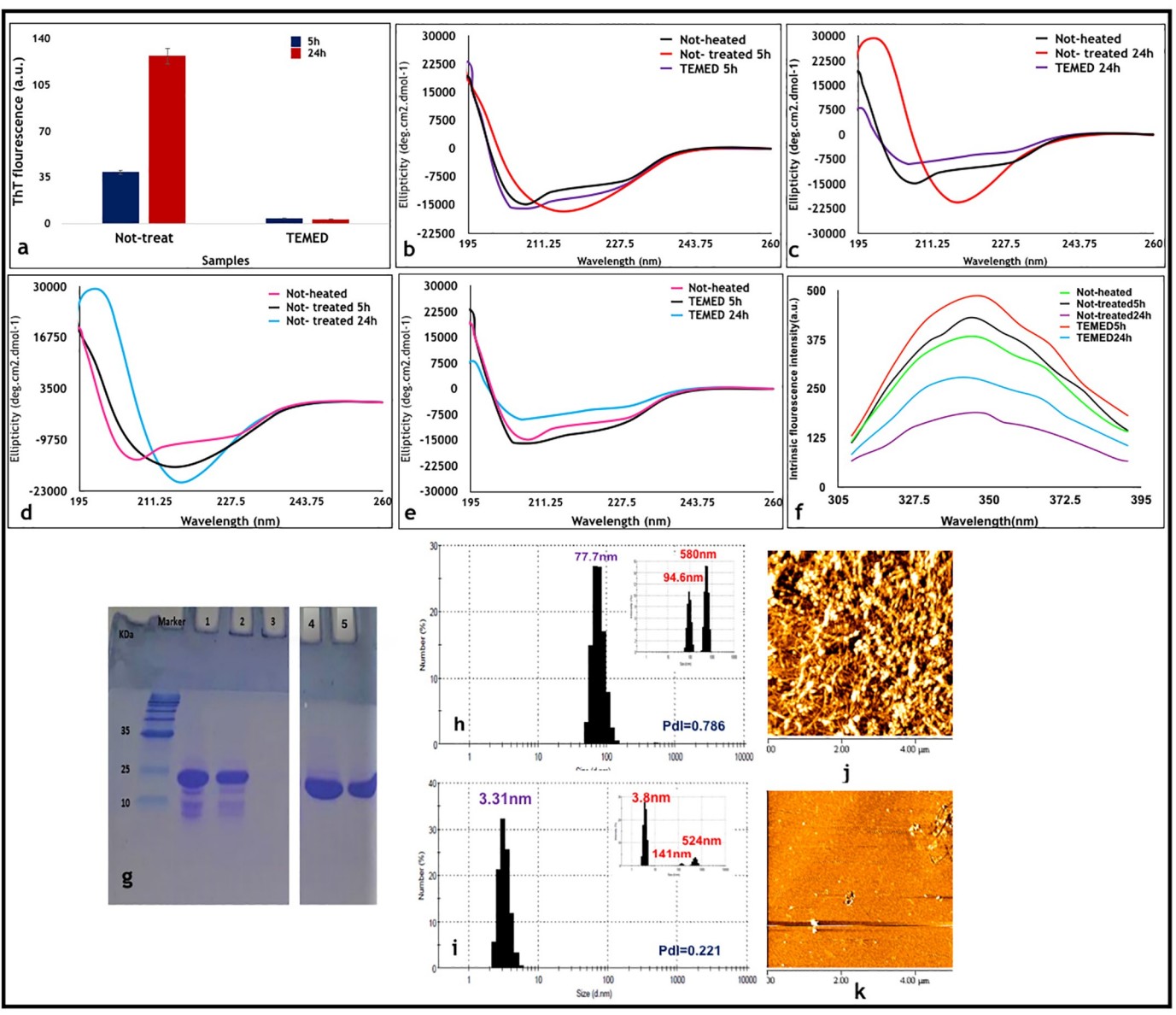

**Fig 1. Characterization of HEWL samples treated with TEMED after 5h and 24h incubation.** (a-e) ThT fluorescence intensities and CD spectra of HEWL in the presence of TEMED in its aroma form after 5h and 24h incubation. (**a**) ThT fluorescence intensities of HEWL Not-treated and treated with aroma of TEMED after 5h and 24h incubation. (**b**) Changes in the secondary structure of HEWL after 5h incubation under fibrillation conditions as monitored by CD. (**c**) Changes in the secondary structure of HEWL after 24h incubation under fibrillation conditions as monitored by CD. (**d, e**) Summary of CD spectra for Not-treated and TEMED-treated HEWL samples, respectively, after 5h and 24h incubation. (**f**) Intrinsic fluorescence analyses of HEWL control and treated samples. Changes in the exposure of the hydrophobic patches of HEWL examined by intrinsic fluorescence spectroscopy are shown. The excitation wavelength was at 280 nm and the fluorescence emission intensity was measured between 300 nm to 400 nm. (**g**) SDS-PAGE analyses of HEWL control and treated samples. The inhibitory effect of aroma from TEMED on HEWL fibrillation as assessed by SDS-PAGE. The wells contained the following: (Marker) Protein marker, (lane 1) Not-heated HEWL, (lane 2) Not-treated5h, (lane 3) Not-treated24h, and (lanes 4 and 5) TEMED5h and TEMED24h, respectively. The entire uncropped gel photo is presented as S6 Fig in the supplementary information section. (**h-k**) DLS and AFM results of HEWL control and aroma treated samples: (**h, j**) Not-treated5h and (**i, k**) TEMED5h. For DLS analysis, all samples were diluted to 1 mg/ml (from 2 mg/ml). The intensity mode of each sample is also provided in the upper right corner of each panel.

formation of HEWL was detected [6] even after 5h incubation [13]. AFM results, confirmed fibril formation in HEWL incubated for 5h in the absence of TEMED (Not-treated5h) (Fig 1j) and DLS results, showed an increase in diameter size to 77.7 nm (Fig 1h), in comparison with Not-heated sample, which we reported previously was 3.36 nm [6]. Following this kinetics

experiment, the effect of TEMED and its ability to prevent fibril formation after 5h was further investigated. Results from CD and ThT fluorescence experiments (Fig 1a–1e) were in line with AFM and DLS results (Fig 1h–1k), confirming that incubation of HEWL with TEMED led to complete inhibition of the fibrillation process even after 24h.

## Intrinsic fluorescence spectroscopy and SDS-PAGE analysis

Using intrinsic fluorescence spectroscopy, we monitored changes in the environment of three different aromatic residues (Trp, Tyr and Phe). As shown in Fig 1f, although TEMED5h had the highest fluorescence intensity, even more than Not-heated HEWL, the Not-treated5h sample in comparison with other samples, especially Not-treated24h, also showed high intrinsic fluorescence intensity indicating that there is a greater exposure of hydrophobic residues after 5h incubation in Not-treated5h, consistent with the presence of a more hydrophobic environment with β-sheet structures [6]. SDS-PAGE analysis showed that while both TEMED5h and TEMED24h samples entered the gel similar to Not-heated sample, the Not-treated24h sample was not able to enter the gel as fibrils were formed and remained in the loading pockets. Appearance of several bands under the main HEWL band in lanes 1 and 2, can be explained to be due to HEWL degradation in acidic pH, which were absent in lanes 4 and 5, as the pH in the presence of TEMED was basic (Fig 1g).

## Thermal stability of HEWL

DSF is a rapid screening method to study the thermal stability of a protein in various conditions. This method is based on the fact that the fluorescence of a protein-binding dye increases with increasing hydrophobicity in the environment. Changes in the exposure of the protein's hydrophobic patches will occur upon heat denaturation. Due to these changes, the dye can interact with exposed hydrophobic regions generated by partially or fully unfolded proteins [14]. Waldron *et al.* 2003, stated that binding a compound to the native state of a protein in the absence of binding to denatured state will stabilize the native state by increasing its thermal stability [15]. Therefore, if TEMED has the ability to bind to the native state of HEWL, then it is expected to increase the melting temperature.

To explore this ability, HEWL samples were prepared in the presence and absence of TEMED (incubated for 5h and 24h under fibrillation conditions) and DSF was used to determine whether or not TEMED resulted in a shift in the $T_m$ value of HEWL. In addition, a second series of samples were prepared in which HEWL was incubated at room temperature for 24h in the presence or absence of different concentrations of TEMED in solution. To find a linear correlation between changes in $T_m$ and an increase in thermal stability, the binding site (s) should be saturated [16]. Therefore, in order to find a logical relationship between the concentration of TEMED and changes in $T_m$ of HEWL, this experiment was done using a wide range of concentrations of TEMED, until the $T_m$ reached a plateau.

As revealed in Fig 2a and 2b, incubation of HEWL under fibrillation conditions in the presence of TEMED, either for 5h or 24h resulted in an increase in the $T_m$ of HEWL by 4 ˚C. On the other hand, as was expected, thermal profiles of HEWL incubated for 2h, without any treatment (Not-treated24h), revealed a non-native structure, interpreted to be related to fibril formation [13].

The results achieved by incubation of the native state of HEWL with different concentrations of TEMED at room temperature showed that by increasing the amount of TEMED in solution, the thermal stability of HEWL was increased (Fig 2c–2f). As shown in Fig 2g, the greatest increase in $T_m$ of HEWL was at a concentration of 200 mM of TEMED. After this concentration (200 mM), higher concentrations of TEMED did not result in higher $T_m$.

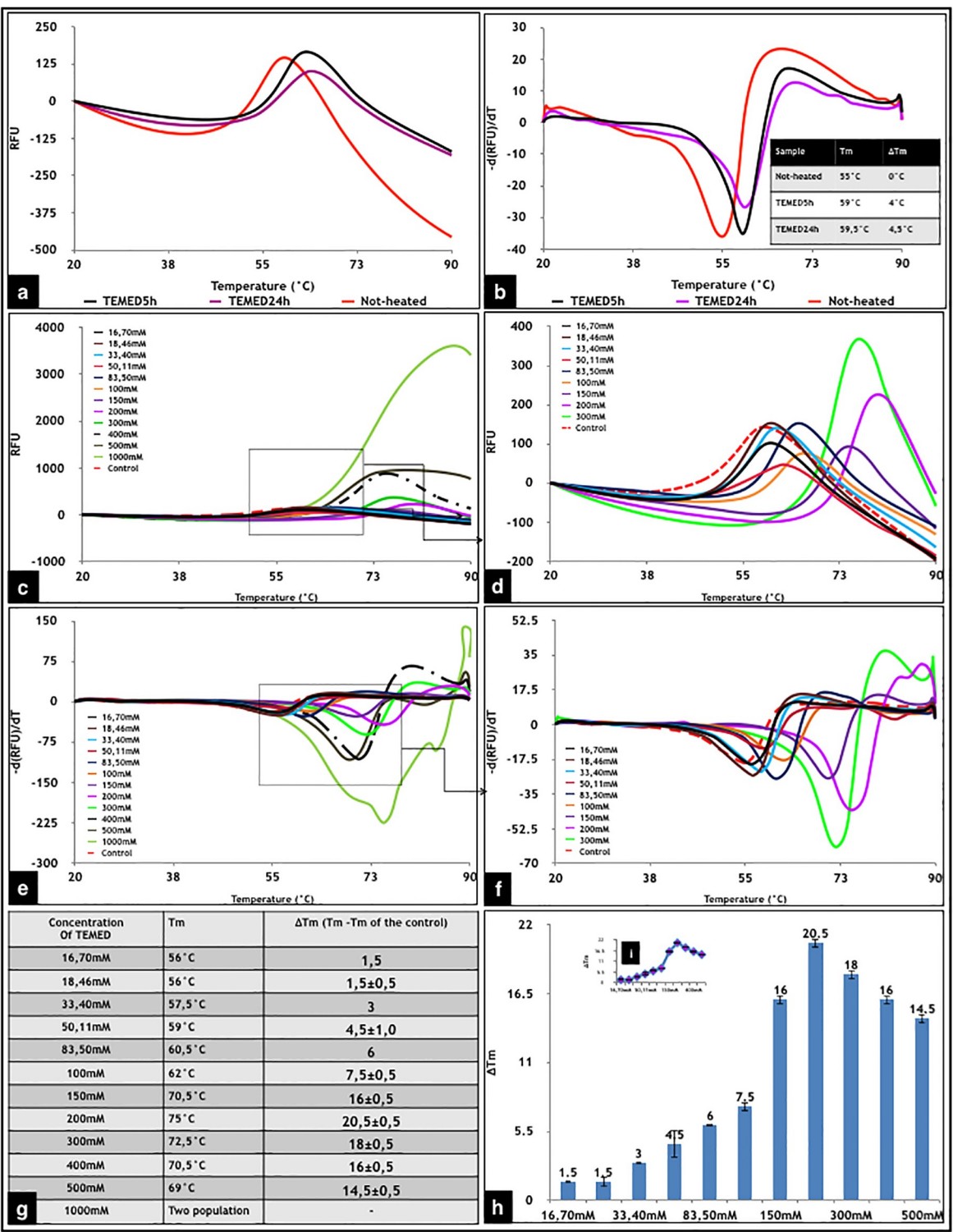

**Fig 2. DSF of HEWL in the absence or presence of TEMED in the aroma form and in solution. (a, b)** HEWL dissolved in 50 mM glycine buffer at pH 2.2 was incubated with TEMED in its aroma form at 54 ˚C under fibrillation conditions for 5h and 24h. **(a)** Thermal melting profile of Not-heated HEWL (control) and HEWL treated with TEMED at 54 ˚C under fibrillation conditions for 5h and 24h. **(b)** First derivative results of Not-heated HEWL (control) and HEWL treated with TEMED at 54 ˚C under fibrillation conditions for 5h and 24h. **(c-h)** DSF of HEWL incubated with different concentrations of TEMED in solution. HEWL dissolved in 50 mM glycine buffer at pH 2.2 was incubated with different concentrations of TEMED at room temperature for 24h. **(c)** Thermal melting profiles of Not-heated

HEWL (control) and HEWL treated with different concentrations of TEMED. (**d**) Magnified version of the thermal melting curve. (**e**) First derivative results of Not-heated HEWL (control) and HEWL treated with different concentrations of TEMED. (**f**) Magnified version of the first derivative results. (**g**) Table representing the influence of the presence of various concentrations of TEMED in changing the $T_m$ of HEWL, achieved by the first derivative results where the $T_m$ of the control sample was 54.5 ˚C. (**h**) 2-D column representation of $\Delta T_m$. (**i**) Increasing trend in $\Delta T_m$ results at different concentrations of TEMED.

Previous studies using TEMED showed changes in the pH of HEWL solution from acidic to basic (from pH 2.2 in glycine buffer to pH 8.3 upon incubation with TEMED in the aroma form). Regarding the importance of pH in this study, as a next step, HEWL was dissolved in glycine buffer pH 8.6 and the DSF experiment was repeated and results compared with that of HEWL dissolved in glycine buffer pH 2.2 (Fig 3).

Correlation between the $T_m$ and pH of HEWL at lower concentrations than 200 mM TEMED (with 200 mM identified as the saturation concentration) was useful in predicting the amount of TEMED in the aroma form used in the fibrillation experiments. Referring back to Fig 2a and 2b, $T_m$ values obtained for TEMED24h and TEMED5h were 59 ˚C and 59.5 ˚C, respectively, at pH of about 8.6. The results obtained from the correlation between $T_m$ and pH using different concentrations of TEMED in solution (Fig 3d), revealed that at $T_m$ of 59 ˚C and pH of 8.6, the concentration of TEMED used was 50.11 mM. Therefore, this suggests that in our fibrillation set-up, the final concentration of TEMED, after incubation under fibrillation conditions at a final pH of 8.3, would have been around 50 mM.

Results in this study showed that the presence of TEMED changed the acidic pH to a basic pH and increased the $T_m$ of HEWL. However, the rise in pH by addition of more TEMED did not necessarily result in higher $T_m$ and greater thermal stability, such that at higher than the optimum concentration of 200 mM TEMED, the $T_m$ started to decrease, which may be due to the saturation level of HEWL (Fig 3c). In other words, although increase in concentration of TEMED and thermal stability of HEWL was correlated with increasing pH of the solution up until 200 mM of TEMED, there was no further linear relationship between the $T_m$ and pH after this concentration.

## Ligand-binding sites

Data collection and refinement statistics of the crystals structures and information about ligand binding to HEWL comprising of TEMED, cryoprotectants including ethylene glycol (EDO) and polyethylene glycol (PGO) and acetate (ACT) from the sodium acetate buffer, are summarised in Tables 1, 2 and 3, respectively. As previously mentioned, TEMED5h and TEMED24h structures refer to HEWL treated under fibrillation conditions with TEMED in the aroma form for 5h and 24h, respectively. Furthermore, as indicated in the DSF results section, based on changes in pH, the final concentration of TEMED at a pH of 8.6, close to the pH of TEMED5h and TEMED24h at 8.3, is estimated at approximately 50 mM. Therefore, the final molar ratio of HEWL:TEMED under these conditions would be around 140 μM: 50 mM or 1:357. On the other hand, TEMED-co, refers to the structure of HEWL co-crystallised with 18.46 mM TEMED, with a molar ratio of 1:131 (HEWL:TEMED). Most of the ligands are associated with HEWL by hydrophobic interactions as shown by the Ligplot analysis (Table 2).

All the different HEWL structures (Table 1) were superposed and presented in three graphical representations (Fig 4). The overall view of each of the structures revealed that four, two and one TEMED attached to HEWL in the structures of TEMED5h, TEMED24h and TEMED-co, respectively.

Structural data show that the active site of HEWL is in the region involving Glu35 and Asp52 [19]. Based on the activity tests from our previous study [6], the greatest increase in

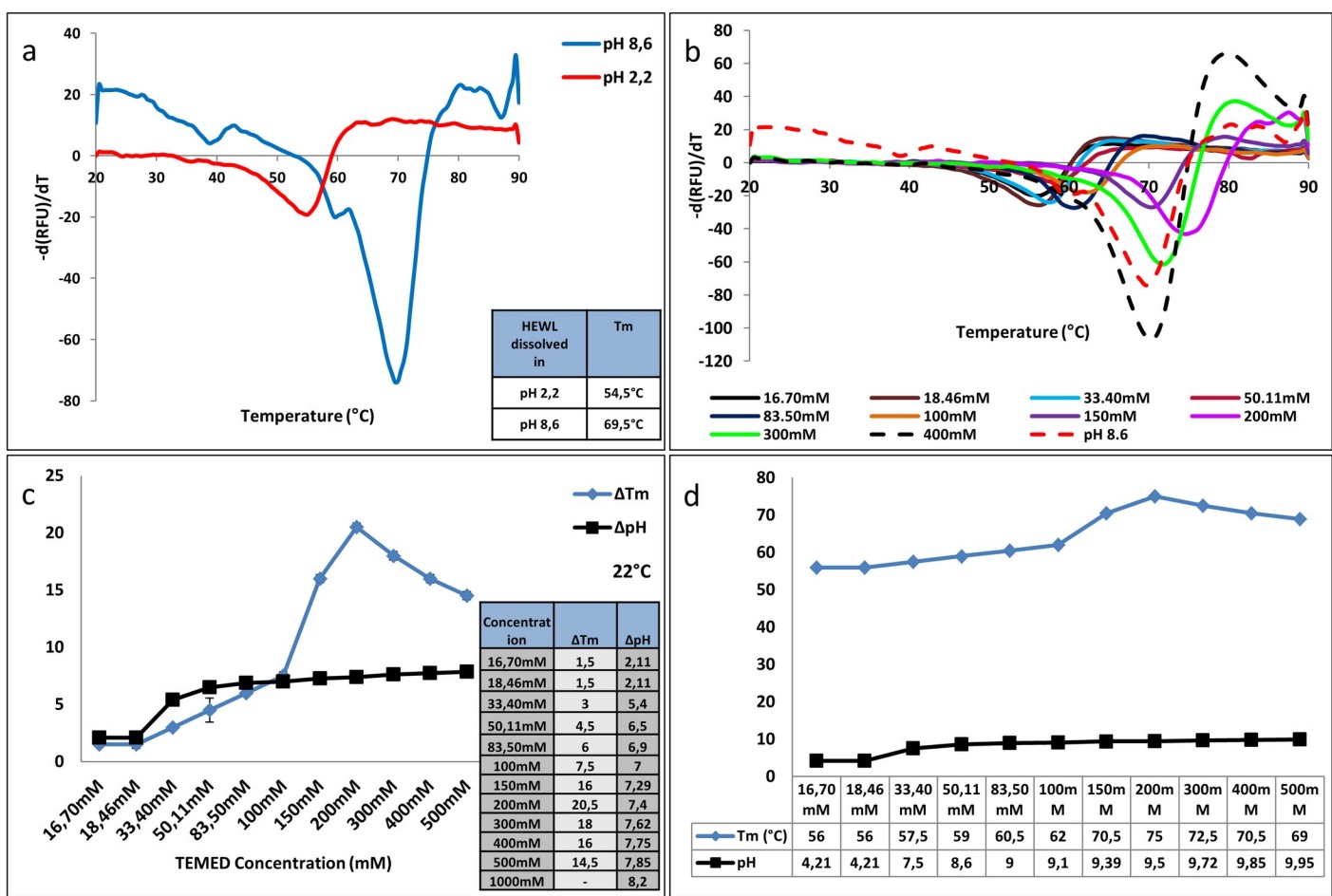

**Fig 3. Correlation between changes in the thermal stability of HEWL and pH of buffer.** (**a**) First derivative results of Not-heated HEWL dissolved in glycine buffer pH 2.2 and pH 8.6. (**b**) First derivative results of Not-heated HEWL dissolved in glycine buffer pH 8.6 in comparison with Not-heated HEWL dissolved in glycine buffer pH 2.2 and incubated with different concentrations of TEMED at room temperature for 24h. (**c**) Relationship between changes in $T_m$ and pH of solution in the presence of different concentrations of TEMED. Table representing the influence of the presence of various concentrations of TEMED in changing the $T_m$ of HEWL, achieved by the first derivative results. The $T_m$ of the control sample was 54.5 ˚C at pH 2.2. The $\Delta T_m$ and $\Delta$pH shows the changes from the control $T_m$ upon addition of increasing amounts of TEMED. (**d**) Relationship between changes in $T_m$ and pH of solution in the presence of different concentrations of TEMED.

enzymatic activity was shown by TEMED24h. This was intriguing since TEMED24h resulted in an increase in HEWL activity, above that of the Not-heated HEWL sample. Now, in light of the structural data obtained, this is understandable. As seen from the Ligplot analysis (Fig 5), in the structure of TEMED5h (Fig 5b), four TEMED molecules bind to HEWL in different positions. One of the TEMED molecules, which was bound near the active site residues including Lys33, Phe34, Glu35 and Asn37, was retained up until 24h in the TEMED24h structure (Fig 5e). The conformation of this TEMED in both TEMED5h and TEMED24h was the same and orientated in a similar way (Fig 5b and 5e). A couple of points can be drawn from these data. Firstly, the similarity in the binding position of TEMED in TEMED5h (Tem2) and TEMED24h (Tem1) shows that the binding of TEMED near the active site of HEWL had a positive effect on the activity of HEWL in comparison to the Not-treated HEWL, which had lost its activity completely, suggesting that TEMED, caused conservation in the activity of HEWL.

**Table 2. Ligand binding sites for TEMED, cryoprotectants and acetate.** The residues involved in either hydrophobic interaction or hydrogen bonding as revealed by the Ligplot+ software version 1.4.5 [17]. Residues with rotamers as revealed from the electron density maps, using Coot from CCP4 package version 2.10.7 [17], are shown in bold.

| PDBs | Ligands | Hydrophobic interactions | H-bond (distance in Å) |
|---|---|---|---|
| **6ABN (pH 8.6)** | EDO | Gln57, Ile58, Trp63, Ile98, Ala107, Trp108 | - |
| | ACT | Asn74, Leu75 | Arg73 (2.76) |
| **6AEA (TEMED5h)** | TEMED1 | Glu35, **Asn44**, Asp52, TEMED2 | - |
| | TEMED2 | Lys33, Phe34, Glu35, Asn37, TEMED1 | - |
| | TEMED3 | Arg5, Ala122, Trp123 | - |
| | TEMED4 | Arg128, **Leu129** | - |
| | PGO1 | Gln57, Ile58, Ile98, Ala107, Trp108 | **Asn59** (2.99) |
| | PGO2 | Trp62, Trp63, Leu75, Asp101 | - |
| **6AD5 (TEMED24h)** | TEMED1 | Lys33, Phe34, Glu35, Asn37, EDO1 | - |
| | TEMED2 | Arg5, Ala122 | - |
| | EDO1 | Glu35, **Asn44**, TEMED1 | - |
| | EDO2 | Gln57, Ile58, Trp63, Ala107, Trp108 | **Asn59** (2.97) |
| **6ADF (TEMED-co)** | TEMED1 | Arg5, Ala122, Trp123 | - |
| | PGO1 | Trp62, Trp63, Leu75, Asp101 | - |
| | PGO2 | Gln57, Ile58, **Asn59**, Trp63, Ala107, Trp108 | - |
| | PGO3 | Asn46, Asp48, Ser50, **Asn59**, Arg61 | - |
| | ACT | Asn74, Leu75 | Arg73 (2.71) |

**Table 3. Binding positions of TEMED in HEWL.**

| Binding hotspots | Ligand binding position | TEMED-co | TEMED5h | TEMED24h |
|---|---|---|---|---|
| Lys33, Phe34, Glu35, Asn37 | A | - | ✓ | ✓ |
| Arg5, Ala122, Trp123 | B | ✓ | ✓ | ✓ |
| Glu35, Asn44, Asp52 | C | - | ✓ | - |
| Arg128, Leu129 | D | - | ✓ | - |

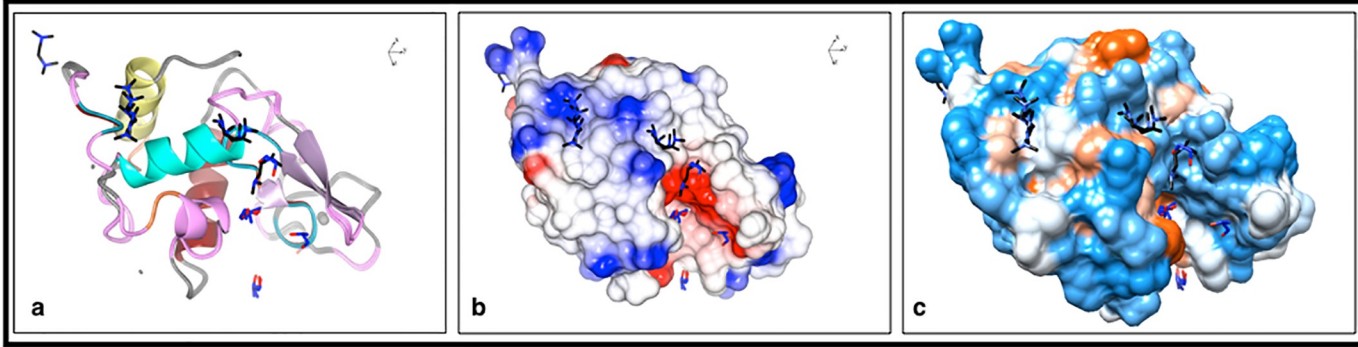

**Fig 4. Three different representations for superposed structures of HEWL with TEMED.** TEMED molecules are coloured black. Oxygen and nitrogen atoms are coloured red and dark blue, respectively. PGO, EDO and ACT are shown as sticks with carbon coloured blue and oxygen coloured red. In addition, the sodium and chloride ions are shown in large and small gray balls, respectively. **(a)** The overall structures of HEWL-TEMED complexes are shown in the ribbon shaped model. **(b)** Electrostatic potential representations of the complexes are shown in their active site views. The structural graphics in panels a and b were generated using CCP4MG version 2.10.7 [18]. **(c)** Hydrophobic surface representations of the structures with TEMED are shown in their active site views. The structure graphics were generated using Chimera version 1.13.1 [12].

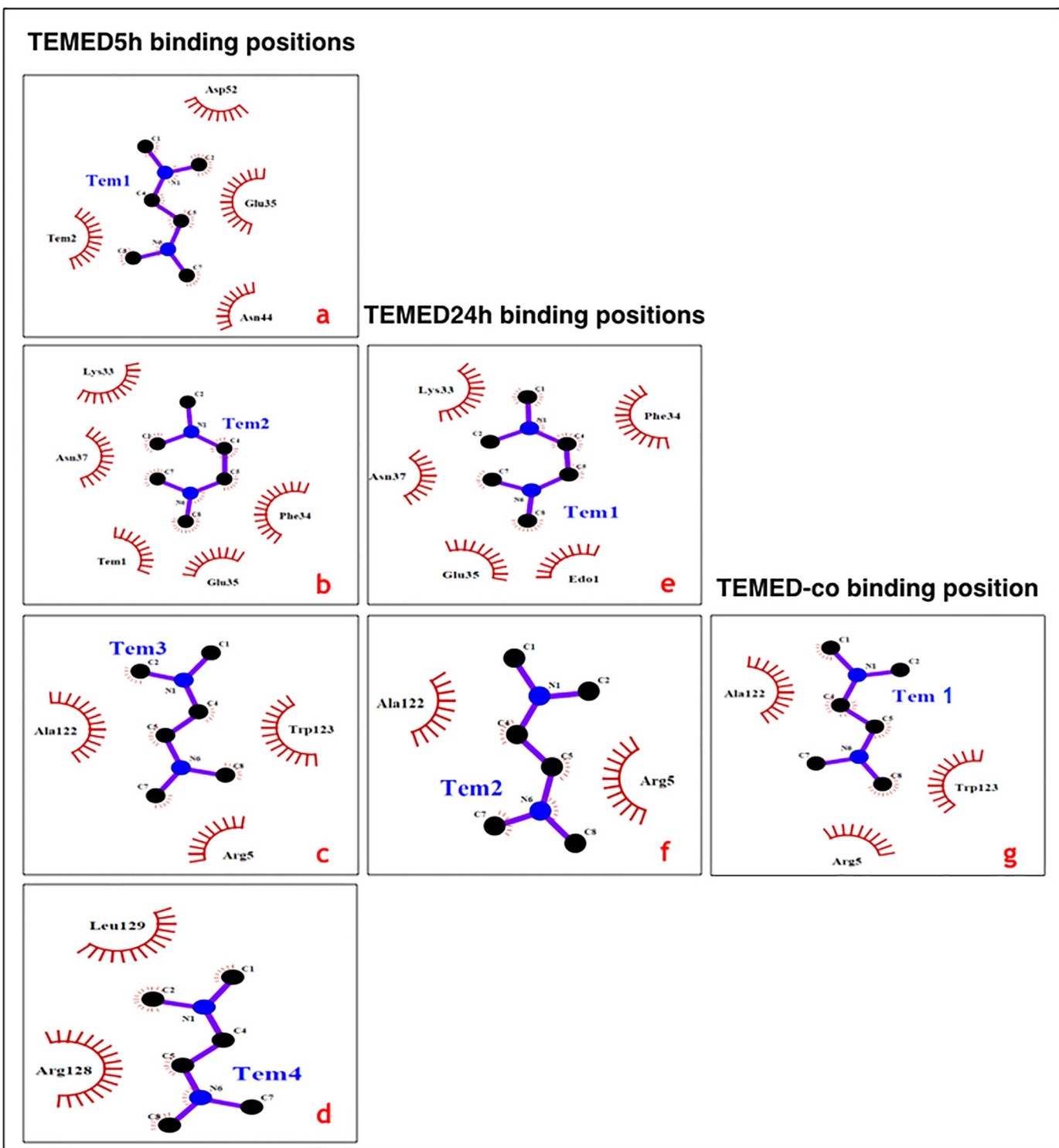

**Fig 5. Binding of TEMED to HEWL as revealed by Ligplot analysis. (a-d)** Binding of four TEMED molecules to HEWL after 5h incubation. **(e and f)** Binding of two TEMED molecules to HEWL after 24h incubation. **(g)** Binding site of a single TEMED to HEWL, upon co-crystallisation. Figure was generated using the Ligplot+ software, version 1.4.5 [11].

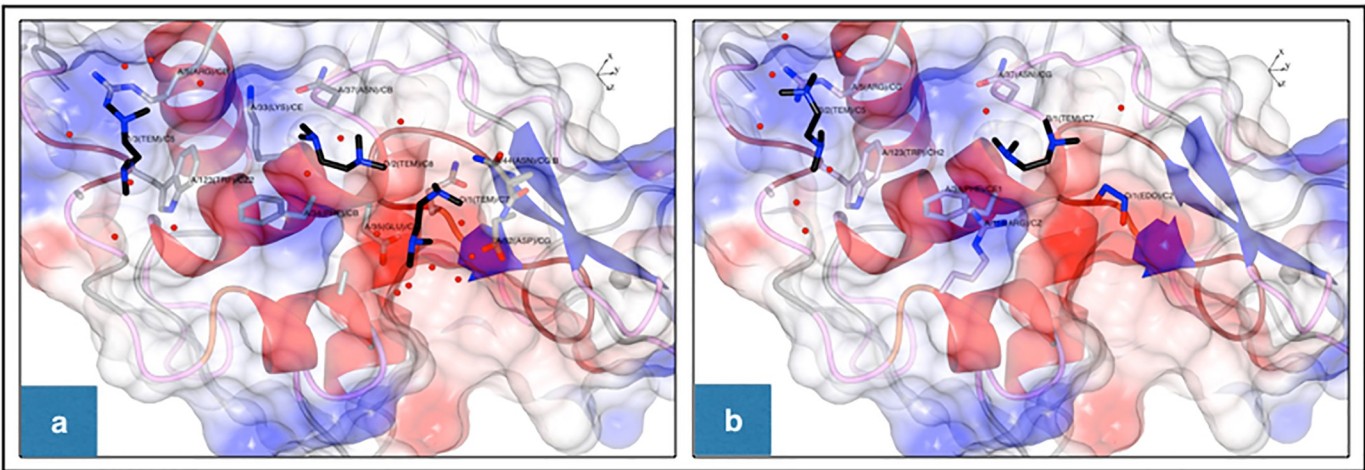

**Fig 6. Binding of TEMED molecules to HEWL. (a)** Three of the four TEMEDs bound to HEWL after 5h incubation. **(b)** Binding of two TEMED molecules to HEWL after 24h incubation. TEMED molecules are coloured black. Oxygen and nitrogen atoms are coloured red and dark blue, respectively. Ethylene glycol (EDO) in TEMED24h is shown as sticks. Water molecules are represented in small red balls. The electrostatic potentials were generated using the Coulombic surface colouring; red is negative and blue is positive. The structural graphics were generated using CCP4MG version 2.10.7 [18].

Secondly, comparing the structures of TEMED5h and TEMED24h revealed that Tem1, which is in the vicinity of Tem2 in TEMED5h (Figs 5a, 5b, 6a and 6b), is no longer visible in the TEMED24h structure. It is suggested that upon further incubation from 5h to 24h, the equivalent of Tem1 in TEMED5h (Figs 5a and 6a) is no longer present in TEMED24h, as the more stable binding position is defined by Tem1 in TEMED24h (Fig 5b and 5e), at a position involving residues Lys33, Phe34, Glu35 and Asn37. There is evidence to suggest that there is a transition in the binding sequence of TEMED from Tem1 to Tem2 in TEMED5h, as Tem2 is associated with a weaker electron density map than Tem1. In TEMED24h, however, Tem1 (equivalent to Tem2 in TEMED5h) is clearly observed at higher contours of the electron density map (Table 3 and S2e, S3 and S4 Figs), although the structure was solved at a lower resolution.

Based on the structural evaluation of the TEMED-HEWL complexes analysed in this study, four main TEMED binding positions (labeled A-D) were identified and reported in Table 3.

In light of the results obtained from this study, amongst the four main positions where TEMED was attached in HEWL (Table 3), position B was occupied with TEMED in all of the structures. In some cases, TEMED molecules were shown to bind weakly in position B, which may raise the question whether their binding should have been reported or not. With regards to the binding of TEMED near Trp123 (position B) in TEMED5h, the intrinsic fluorescence results showed that there was no quenching whatsoever at this position and instead an increased intrinsic fluorescence intensity was observed in comparison to the Not-heated HEWL (Fig 1f). This may be explained by the increased hydrophobicity in the environment of Trp123 due to the presence of TEMED, a hydrophobic compound (log P = 0.30), which supports the structural data in this study. Therefore, based on both intrinsic fluorescence studies and the existence of multiple rotamers and changes in B factors of certain residues (S2 and S3 Tables, S1 Fig), it was plausible to confirm binding in position B. In the case of positions C and D, a TEMED molecule was observed in TEMED5h, however after 24h incubation, TEMED was not observed in this position in the structure of TEMED24h.

Additionally, according to the results obtained in this study, position A could be strongly suggested as the "hotspot" for inhibition of fibril formation in HEWL. This position is near the

active site of HEWL and binding of TEMED to this position could explain why treatment of HEWL with TEMED resulted in an increased enzymatic activity. Harada *et al.* 2007, claimed that existence of lysyl residues with a positive charge at the C-terminal of α-helix A, α-helix B and α-helix C (consisting of residues 5–15, 26–36 and 89–100, respectively) contributed to the stability of helices, resulting in conformational stability of HEWL [20]. Additionally, Kawamura *et al.* 2008, stated that Arg114 and Phe34 are two residues which are responsible for the productive binding at right-sided binding site of HEWL and that the CH-π interaction between the guanidyl side chain $(C(NH_2)_2)$ of Arg114 and the phenyl ring of Phe34 on the molecular surface could stabilize the Arg114 side chain and position it to accept the substrate molecule [21]. Accordingly, the interaction between Arg114 and Phe34 is suggested to be the important structural factor in maintaining the native conformation of HEWL. Based on their results, Arg114 in HEWL is thought to play an important role in lysozyme catalysis, especially in transglycosylation. Now looking at our results of the enzymatic activity test and the structure of HEWL treated with TEMED, the presence of TEMED near Phe34 (with hydrophobic interactions) could have an effect on the stabilization of Arg114. Additionally, it is plausible to say that the CH-π interaction between the methylamine $(N(CH_3)_2)$ groups in TEMED (of which there are two) and phenyl ring in Phe34 (providing the π system), in addition to the CH-π interaction between the guanidyl side chain of Arg114 and again that of the phenyl ring of Phe34, are the main contributors in enhancing the ability of HEWL to accept its substrate, such that we see greater enzymatic activity in HEWL in the presence of TEMED and greater intrinsic fluorescence emission.

## Conclusions

In this study, TEMED was chosen because of its pungent fishlike odour and its similarity to putrescence and cadaverine as polyamines, also known as the "smell of death" [3]. A common route for polyamine formation is decarboxylation of precursor amino acids by enzymes of bacterial origin. It is revealed that polyamines regulate growth in pathogens and can therefore have an influence on infection and parasitic disease [22]. As such, bacteria, during growth and activity make more and more polyamines. As we can see from the structural results of this study, TEMED binds to HEWL (as an enzyme that lyses the cell wall of bacteria) at a hotspot comprising residues Lys33, Phe34, Glu35 and Asn37 providing complete inhibition of fibril formation, greater enzymatic activity and increased thermal stability. This may suggest that TEMED in its aroma form acts as a catalyst of HEWL activity (as an antibacterial enzyme), which is in line with a report which emphasizes on the importance of smell and shows how bacteria may use the sense of smell for communication [23]. Furthermore, changes in pH caused by TEMED, even in the aroma form, suggests that bacterial infection may result in a similar pH as a consequence of protein degradation products, including putrescence and cadaverine, and provide an activation signal for HEWL, which confirms the importance of the smell or aroma of a compound, usually a small molecule, as a signal affecting the structure and function of proteins, such as HEWL investigated in this study. On the other hand, as HEWL is used as a model protein for fibril formation, this work also shows that attempts to interfere with fibrillation processes may also find fruitful grounds in the identification of small molecules that can stabilize the native structure of proteins known to undergo fibrillation.

## Supporting information

**S1 Fig. Shifts or rotamers of Arg128 and Asn19 in the different structures. (a,e)** HEWL at pH 8.6 (PDB ID 6ABN), **(b,f)** TEMED-co (PDB ID 6ADF), **(c,g)** TEMED5h (PDB ID 6AEA) and **(d,h)** TEMED24h (PDB ID 6AD5). Structure graphics were generated using Coot from

CCP4 package version 2.10.7 [17].
(DOCX)

**S2 Fig. Electron density maps showing binding of TEMED to HEWL. (a-d)** Binding of four TEMED molecules to HEWL after 5h incubation. (**e and f**) Binding of two TEMED molecules to HEWL after 24h incubation. (**g**) Binding site of a single TEMED in HEWL upon co-crystallisation. The σA-weighted 2Fobs−Fcalc maps were contoured at 0.5 sigma and generated in Coot from CCP4 package version 2.10.7 [17].
(DOCX)

**S3 Fig. Electron density map showing binding of TEMED to HEWL after 5h incubation.** The σA-weighted 2Fobs−Fcalc maps (in blue mesh) were contoured at 0.5 and 1.0 sigma levels and the σA-weighted Fobs-Fcalc maps (in green and red mesh) were contoured at a default sigma level of 3.0. The maps were generated in Coot from CCP4 package version 2.10.7 [17].
(DOCX)

**S4 Fig. Electron density map showing binding of TEMED to HEWL after 24h incubation.** The σA-weighted 2Fobs−Fcalc maps (in blue mesh) were contoured at 0.5 and 1.0 sigma levels and the σA-weighted Fobs-Fcalc maps (in green and red mesh) were contoured at a default sigma level of 3.0. The maps were generated in Coot from CCP4 package version 2.10.7 [17].
(DOCX)

**S5 Fig. Electron density map showing binding of TEMED to HEWL upon co-crystallisation.** The σA-weighted 2Fobs−Fcalc maps (in blue mesh) were contoured at 0.5 and 1.0 sigma levels and the σA-weighted Fobs-Fcalc maps (in green and red mesh) were contoured at a default sigma level of 3.0. The maps were generated in Coot from CCP4 package version 2.10.7 [17].
(DOCX)

**S6 Fig. The inhibitory effect of aroma from TEMED on HEWL fibrillation as assessed by SDS-PAGE.** The wells contain the following: (Marker) Protein marker, (lane 1) Not-heated HEWL, (lane 2) Not-treated5h, (lane 3) Not-treated24h, and (lanes 8 and 9) TEMED5h and TEMED24h, respectively. Lanes 4–7 contain samples not relevant to this research article. This Supplementary Figure is provided since the original uncropped and unadjusted images underlying all blot or gel results are to be reported.
(DOCX)

**S1 Table. Various concentrations of TEMED used in this study using a vapour diffusion method.**
(DOCX)

**S2 Table. Different rotamers of residues in HEWL in complex with TEMED and at pH 8.6.**
(DOCX)

**S3 Table. B-Factor values for Arg14, Glu7 and His15, Asp18, Asn19 and Arg128 in the structures of HEWL in complex with TEMED and at pH 8.6.**
(DOCX)

**S1 Data.**
(ZIP)

## Acknowledgments

We thank the ALBA synchrotron for the use of the XALOC beamline and the ALBA BioLab for hosting Miss Zahra Seraj during her two months of OPEN SESAME fellowship training program. We thank Dr Xavi Carpena for mentoring Miss Zahra Seraj, Dr Fernando Gil Ortiz for assistance with some of the X-ray diffraction data collections and Dr Roeland Boer as the responsible scientist of the XALOC beam line at ALBA Synchrotron. We are grateful to Albert Castellvi for helping in fishing some of the crystals and giving advice in using the XDS and Scala software and Rick Oerlemans for helping in the DSF experiments and advice in data processing. We thank Professor Alexander Dömling as the chair of the Drug Design Department for accepting Miss Zahra Seraj to take up her student placement at the University of Groningen, Netherlands. We also thank the structural biology lab of the drug design group for hosting Miss Zahra Seraj for her six months student placement. We are also thankful to Mr Fouad Mehraban for accompanying us for the data collection trip to ALBA synchrotron.

## Author Contributions

**Conceptualization:** Arefeh Seyedarabi.

**Data curation:** Zahra Seraj, Matthew R. Groves.

**Formal analysis:** Zahra Seraj, Matthew R. Groves, Arefeh Seyedarabi.

**Funding acquisition:** Shahin Ahmadian, Arefeh Seyedarabi.

**Investigation:** Zahra Seraj, Matthew R. Groves, Arefeh Seyedarabi.

**Methodology:** Zahra Seraj, Matthew R. Groves, Arefeh Seyedarabi.

**Project administration:** Arefeh Seyedarabi.

**Resources:** Shahin Ahmadian, Matthew R. Groves, Arefeh Seyedarabi.

**Supervision:** Matthew R. Groves, Arefeh Seyedarabi.

**Validation:** Arefeh Seyedarabi.

**Writing – original draft:** Zahra Seraj.

**Writing – review & editing:** Matthew R. Groves, Arefeh Seyedarabi.

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
