## [Decision Letter · Decision Letter 0]

15 Jan 2020

PONE-D-19-32635

The aroma of TEMED as an activation and stabilizing signal for the antibacterial enzyme HEWL

PLOS ONE

Dear Dr Seyedarabi,

Thank you for submitting your manuscript to PLOS ONE. After careful consideration, we feel that it has merit but does not fully meet PLOS ONE’s publication criteria as it currently stands. Therefore, we invite you to submit a revised version of the manuscript that addresses the points raised during the review process.

We would appreciate receiving your revised manuscript by Feb 29 2020 11:59PM. To enhance the reproducibility of your results, we recommend that if applicable you deposit your laboratory protocols in protocols.io, where a protocol can be assigned its own identifier (DOI) such that it can be cited independently in the future. For instructions see: http://journals.plos.org/plosone/s/submission-guidelines#loc-laboratory-protocols

We look forward to receiving your revised manuscript.

Kind regards,

Yong-Bin Yan, Ph.D.

Academic Editor

PLOS ONE

Journal Requirements:

"We acknowledge financial support of the Presidential Deputy for Science and Technology, the

Institute for Research in Fundamental Sciences and the Iranian Light Source Facility for the

opportunity to be users at the ALBA synchrotron program 2017 (proposal number

ID2017062245). We acknowledge financial support of the OPEN SESAME for the two months

student training fellowship for Miss Zahra Seraj (2018) and the Ministry of Science and Technology

of Iran for six months research placement of Miss Zahra Seraj at University of Groningen,

Netherlands."

3. We note that Figure 8 in your submission contain copyrighted images. All PLOS content is published under the Creative Commons Attribution License (CC BY 4.0), which means that the manuscript, images, and Supporting Information files will be freely available online, and any third party is permitted to access, download, copy, distribute, and use these materials in any way, even commercially, with proper attribution. For more information, see our copyright guidelines: http://journals.plos.org/plosone/s/licenses-and-copyright.

a)     You may seek permission from the original copyright holder of Figure(s) [#] to publish the content specifically under the CC BY 4.0 license.

Reviewers' comments:

Reviewer's Responses to Questions

**Comments to the Author**

1. Is the manuscript technically sound, and do the data support the conclusions?

Reviewer #1: Partly

Reviewer #2: Partly

2. Has the statistical analysis been performed appropriately and rigorously? 

Reviewer #1: No

Reviewer #2: Yes

3. Have the authors made all data underlying the findings in their manuscript fully available?

Reviewer #1: Yes

Reviewer #2: Yes

4. Is the manuscript presented in an intelligible fashion and written in standard English?

Reviewer #1: No

Reviewer #2: No

5. Review Comments to the Author

Reviewer #1: Comments:

It is interesting to reveal the effect of unpleasant smell such as TEMED on the structure and function of proteins. The work by Seraj et al. describes TEMED could inhibit the fibril formation of Hen egg white lysozyme (HEWL). Further, they identified the binding site(s) of TEMED and HEWL, and the hotspot(s) required to inhibit fibril formation. In addition, the binding of TEMED and HEWL increased the activity and thermal stability of HEWL. In conclusion, the role of the "smell of death" could serve as an important signal molecule to affect the activity and stability of HEWL. I think the work may be interesting for the potential readers of the journal. However, the following issues should be clearly addressed before it is accepted for publication.

1. The researchers identified the binding site(s) of TEMED and HEWL, and the hotspot(s) required to inhibit fibril formation through the Ligplot analysis. Could you please kindly provide more methods/techniques such as site-directed mutagenesis to confirm your findings?

2. Is it specific for TEMED to inhibit fibril formation and increase the activity and thermal stability of HEWL? Does it affect the structure and activity of other proteins such as BSA?

3. Is there other diamine affecting the activity and structure of HEWL?

4. Could you explain why the residues Lys33, Phe34, Glu35 and Asn37 are important for the binding of TEMED and HEWL?

5. The abstract should be revised carefully to describe the goal of this study, the main findings and the conclusion to help readers better understand the novelty of this study.

6. The figures and tables in this manuscript could not meet the science criterion and should be thoroughly reorganized and polished to present to the readers.

7. The conclusion is too long. It should be concise and highlight the most important findings and scientific significance. Please revise it carefully.

8. The authors should have their work reviewed by a proper translation/reviewing service or native-English speaker before submission. The manuscript needs careful editing by someone with particular attentions to English grammar, spelling, and sentence structure so that the goal and results of the study are clear to the readers.

In my opinion, the article in the present form is not suitable for publication in PLoS One, and should undergo a round of major revision before publication at least.

Reviewer #2: This study tried to study how "unpleasant smell" molecules (using TEMED as a model molecule) can be signal molecules and provide a structural mechanism. However, the writing of the paper is not clear even English is good. There are two major issues for this manuscript:

1. HEWL is more like a random model protein the authors chose to study how TEMED affects protein stability and fiber formation. There is no clear explanation why interaction of TEMED with HEWL will have a biological effect. Using model proteins to study impact of TEMED is acceptable, but making farfetched biological functions is not, and can be misleading.

2. Although several crystal structures were determined in complex with TEMED, the omit maps (so called Fo-Fc) and 2Fo-Fc are not provided. The authors have to provide these two maps to show that the TEMED molecules are really in the complex, instead of artifacts. For a small molecule like TEMED, arbitrarily placing atoms is not likely to impact Rfree or R in a noticeable way. By checking the B factors of some pdb structures published in this study, some TEMED molecules have high B factors, indicating low occupancy or improper model fitting.

6. PLOS authors have the option to publish the peer review history of their article (what does this mean?). If published, this will include your full peer review and any attached files.

Reviewer #1: No

Reviewer #2: No

---

## [Author Response · Author response to Decision Letter 0]

2 Apr 2020

Please see the file uploaded

Reviewer #1: Comments:

It is interesting to reveal the effect of unpleasant smell such as TEMED on the structure and function of proteins. The work by Seraj et al. describes TEMED could inhibit the fibril formation of Hen egg white lysozyme (HEWL). Further, they identified the binding site(s) of TEMED and HEWL, and the hotspot(s) required to inhibit fibril formation. In addition, the binding of TEMED and HEWL increased the activity and thermal stability of HEWL. In conclusion, the role of the "smell of death" could serve as an important signal molecule to affect the activity and stability of HEWL. I think the work may be interesting for the potential readers of the journal. However, the following issues should be clearly addressed before it is accepted for publication.

1. The researchers identified the binding site(s) of TEMED and HEWL, and the hotspot(s) required to inhibit fibril formation through the Ligplot analysis. Could you please kindly provide more methods/techniques such as site-directed mutagenesis to confirm your findings?

Answer: Thank you for your suggestion. However, in this study, we used X-ray crystallography with high resolution crystal structures to identify the binding site(s) and in particular the hotspot for inhibition of fibril formation in HEWL. The Ligplot images are a means to present the data obtained from our crystal structures showing residues in proximity to TEMED, their distances and types of interactions. 

Additionally, as mentioned in our manuscript, the binding of TEMED increased the activity of HEWL further emphasising the position and residues involved in binding:

Additionally, according to the results obtained in this study, position A could be strongly suggested as the "hotspot" for inhibition of fibril formation in HEWL. This position is near the active site of HEWL and binding of TEMED to this position could explain why treatment of HEWL with TEMED resulted in an increased enzymatic activity. Harada et al. 2007, claimed that existence of lysyl residues with a positive charge at the C-terminal of α-helix A, α-helix B and α-helix C (consisting of residues 5-15, 26-36 and 89-100, respectively) contributed to the stability of helices, resulting in conformational stability of HEWL [1]. Kawamura et al. 2008, stated that Arg114 and Phe34 are two residues which are responsible for the productive binding at right-sided binding site of HEWL and that the CH-π interaction between the guanidyl side chain (C(NH2)2) of Arg114 and the phenyl ring of Phe34 on the molecular surface could stabilize the Arg114 side chain and fix it to accept the substrate molecule via the right-sided mode [2]. Accordingly, the interaction between Arg114 and Phe34 is suggested to be the important structural factor in maintaining the native conformation of HEWL. Based on their results, Arg114 in HEWL is thought to play an important role in lysozyme catalysis, especially in transglycosylation. Now looking at our results of the enzymatic activity test and the structure of HEWL treated with TEMED, the presence of TEMED near Phe34 (with hydrophobic interactions) could have an effect on the stabilization of Arg114. Additionally, it is plausible to say that the CH-π interaction between the methylamine (N(CH₃)₂) groups in TEMED (of which there are two) and phenyl ring in Phe34 (providing the π system), in addition to the CH-π interaction between the guanidyl side chain of Arg114 and again that of the phenyl ring of Phe34, are the main contributors in enhancing the ability of HEWL to accept its substrate, such that we see greater enzymatic activity in HEWL in the presence of TEMED and greater intrinsic fluorescence emission.

2. Is it specific for TEMED to inhibit fibril formation and increase the activity and thermal stability of HEWL? Does it affect the structure and activity of other proteins such as BSA?

Answer: Since the aim of this research was to use HEWL as a model protein for fibrillation studies, we were not concerned with the effect of TEMED on other proteins, but rather the effect of different small molecules on inhibition of HEWL fibril formation. However, we have preliminary DSF results showing that the presence of different concentrations of TEMED for two different purified proteins, Thiosulfate transferase (hTst) and 1-deoxy-D-xylulose 5-phosphate synthase from Mycobacterium tuberculosis (mtDXS), either showed no thermal stability changes or resulted in decreased Tm, respectively (please see Figure below).

Therefore, it appears so far that TEMED is specific for increasing the thermal stability of HEWL. 

The DSF results are provided to answer the respectable reviewer’s question, but is preliminary and preferably not to be included in this manuscript.

Additionally, Anthony et al., reported from their study that in contrast to the effects of TEMED, which we observed on HEWL, all the three similar polyamines listed in their study including putrescine, spermidine and spermine resulted in an increased alpha synuclein aggregation [3]. Another report from Luo et al. showed that cellular polyamines promote amyloid-Beta (Aβ) peptide fibrillation [4]. Furthermore, it is reported that the acceleration of aggregation by certain polyamines suggest that they may also facilitate neurodegeneration [3].

DSF results of 1-deoxy-D-xylulose 5-phosphate synthase from Mycobacterium tuberculosis (mtDXS) and Thiosulfate transferase (hTst) in either the absence or presence of different concentrations of TEMED in solution. (a) 1-deoxy-D-xylulose 5-phosphate synthase from Mycobacterium tuberculosis (mtDXS). (b) Thiosulfate transferase (hTst).

3. Is there other diamine affecting the activity and structure of HEWL?

Answer: Yes, there is a report from Okanojo, et al., at 2005 which showed that diamines prevent thermal aggregation and inactivation of lysozyme [5].

4. Could you explain why the residues Lys33, Phe34, Glu35 and Asn37 are important for the binding of TEMED and HEWL?

Answer: As we mentioned in our manuscript, Kawamura et al. 2008 [6], stated that Arg114 and Phe34 are two residues which are responsible for the productive binding at right-sided binding site of HEWL. Furthermore, the CH-Π interaction between the guanidyl side chain (C(NH2)2) of Arg114 and the phenyl ring of Phe34 on the molecular surface could stabilize the Arg114 side chain and fix it to accept the substrate molecule via the right-sided mode. Accordingly, the interaction between Arg114 and Phe34 is suggested to be the important structural factor in maintaining the proper conformation of HEWL. Based on their results, Arg114 in HEWL is thought to play an important role in lysozyme catalysis, especially in transglycosylation. Now looking at our results of the enzymatic activity test and the structure of HEWL treated with TEMED, the presence of TEMED near Phe34 (with hydrophobic interactions) could have an effect on the stabilization of Arg114. Additionally, it is plausible to say that the CH-Π interaction between the methylamine (N(CH₃)₂) groups in TEMED (of which there are two) and phenyl ring in Phe34 (providing the Π system) in addition to the CH-Π interaction between the guanidyl side chain of Arg114 and again that of the phenyl ring of Phe34, is the main contributor in enhancing the ability of HEWL to accept its substrate such that we see greater enzymatic activity in HEWL in the presence of TEMED and greater intrinsic fluorescence emission.

Additionally, binding of TEMED close to Glu35 as one of the residues in the active site including Glu35 and Asp52, which resulted in increasing in the activity of HEWL, was supportive data about the importance of this position.

5. The abstract should be revised carefully to describe the goal of this study, the main findings and the conclusion to help readers better understand the novelty of this study.

Answer: Thank you. The abstract has been improved.

6. The figures and tables in this manuscript could not meet the science criterion and should be thoroughly reorganized and polished to present to the readers.

Answer: Changes were applied as much as possible.

7. The conclusion is too long. It should be concise and highlight the most important findings and scientific significance. Please revise it carefully.

Answer: Changes were applied to the conclusion section of the manuscript.

8. The authors should have their work reviewed by a proper translation/reviewing service or native-English speaker before submission. The manuscript needs careful editing by someone with particular attentions to English grammar, spelling, and sentence structure so that the goal and results of the study are clear to the readers.

Answer: Thank you for your suggestion. This was done.

In my opinion, the article in the present form is not suitable for publication in PLoS One, and should undergo a round of major revision before publication at least.

Reviewer #2:

This study tried to study how "unpleasant smell" molecules (using TEMED as a model molecule) can be signal molecules and provide a structural mechanism. However, the writing of the paper is not clear even English is good. There are two major issues for this manuscript:

1. HEWL is more like a random model protein the authors chose to study how TEMED affects protein stability and fiber formation. There is no clear explanation why interaction of TEMED with HEWL will have a biological effect. Using model proteins to study impact of TEMED is acceptable, but making farfetched biological functions is not, and can be misleading.

Answer: As we explained in this study, TEMED was chosen because of its pungent fishlike odour and its similarity to putrescence and cadaverine as polyamines, also known as the "smell of death”, which are produced by decarboxylation of precursor amino acids by enzymes of bacterial origin. On the other hand, based on our previous study, it was interesting for us that fibril formation in HEWL, as an antibacterial enzyme, was inhibited by TEMED with properties mentioned above. Therefore, we decided to extend our study and tried to find whether or not these results were related to any potential biological effects or roles. Interesting we found, using results from a number of techniques, that TEMED was a specific ligand for HEWL.

2. Although several crystal structures were determined in complex with TEMED, the omit maps (so called Fo-Fc) and 2Fo-Fc are not provided. The authors have to provide these two maps to show that the TEMED molecules are really in the complex, instead of artifacts. For a small molecule like TEMED, arbitrarily placing atoms is not likely to impact Rfree or R in a noticeable way. By checking the B factors of some pdb structures published in this study, some TEMED molecules have high B factors, indicating low occupancy or improper model fitting.

Answer: Omit maps are now provided in the supplementary information Figures S2-S5.

Bibliography

1. Harada, A., et al., Relationship between the stability of Hen Egg-White lysozymes mutated at sites designed to interact with α-helix dipoles and their secretion amounts in yeast. Bioscience, biotechnology, and biochemistry, 2007. 71(12): p. 2952-2961.

2. Kawamura, S., et al., The role of Arg114 at subsites E and F in reactions catalyzed by hen egg-white lysozyme. Bioscience, biotechnology, and biochemistry, 2008. 72(3): p. 823-832.

3. Antony, T., et al., Cellular polyamines promote the aggregation of alpha-synuclein. J Biol Chem, 2003. 278(5): p. 3235-40.

4. Luo, J., et al., Cellular polyamines promote amyloid-beta (Abeta) peptide fibrillation and modulate the aggregation pathways. ACS Chem Neurosci, 2013. 4(3): p. 454-62.

5. Okanojo, M., et al., Diamines prevent thermal aggregation and inactivation of lysozyme. J Biosci Bioeng, 2005. 100(5): p. 556-61.

6. Kawamura, S., et al., The role of Arg114 at subsites E and F in reactions catalyzed by hen egg-white lysozyme. Biosci Biotechnol Biochem, 2008. 72(3): p. 823-32.

---

## [Decision Letter · Decision Letter 1]

27 Apr 2020

The aroma of TEMED as an activation and stabilizing signal for the antibacterial enzyme HEWL

PONE-D-19-32635R1

Dear Dr. Seyedarabi,

We are pleased to inform you that your manuscript has been judged scientifically suitable for publication and will be formally accepted for publication once it complies with all outstanding technical requirements.

With kind regards,

Yong-Bin Yan, Ph.D.

Academic Editor

PLOS ONE

Additional Editor Comments (optional):

Reviewers' comments:

Reviewer's Responses to Questions

**Comments to the Author**

1. If the authors have adequately addressed your comments raised in a previous round of review and you feel that this manuscript is now acceptable for publication, you may indicate that here to bypass the “Comments to the Author” section, enter your conflict of interest statement in the “Confidential to Editor” section, and submit your "Accept" recommendation.

Reviewer #1: All comments have been addressed

2. Is the manuscript technically sound, and do the data support the conclusions?

Reviewer #1: Yes

3. Has the statistical analysis been performed appropriately and rigorously? 

Reviewer #1: Yes

4. Have the authors made all data underlying the findings in their manuscript fully available?

Reviewer #1: Yes

5. Is the manuscript presented in an intelligible fashion and written in standard English?

Reviewer #1: Yes

6. Review Comments to the Author

Reviewer #1: (No Response)

7. PLOS authors have the option to publish the peer review history of their article (what does this mean?). If published, this will include your full peer review and any attached files.

Reviewer #1: No

---

## [Editor Report · Acceptance letter]

8 May 2020

PONE-D-19-32635R1 

The aroma of TEMED as an activation and stabilizing signal for the antibacterial enzyme HEWL 

Dear Dr. Seyedarabi:

I am pleased to inform you that your manuscript has been deemed suitable for publication in PLOS ONE. Congratulations! Your manuscript is now with our production department. 

With kind regards,

on behalf of

Dr. Yong-Bin Yan 

Academic Editor

PLOS ONE